# Polar mesospheric ozone loss initiates downward coupling of solar signal in the Northern Hemisphere

Annika Seppälä [1] ✉, Niilo Kalakoski [2], Pekka T. Verronen [2,3], Daniel R. Marsh [4,5], Alexey Yu. Karpechko[6] & Monika E. Szelag [2]

Solar driven energetic particle precipitation (EPP) is an important factor in polar atmospheric ozone balance and has been linked to ground-level regional climate variability. However, the linking mechanism has remained ambiguous. The observed and simulated ground-level changes start well before the processes from the main candidate, the so-called EPP-indirect effect, would start. Here we show that initial reduction of polar mesospheric ozone and the resulting change in atmospheric heating rapidly couples to dynamics, transferring the signal downwards, shifting the tropospheric jet polewards. This pathway is not constrained to the polar vortex. Rather, a subtropical route initiated by a changing wind shear plays a key role. Our results show that the signal propagates downwards in timescales consistent with observed tropospheric level climatic changes linked to EPP. This pathway, from mesospheric ozone to regional climate, is independent of the EPP-indirect effect, and solves the long-standing mechanism problem for EPP effects on climate.

Energetic particle precipitation (EPP) is natural solar forcing into the atmosphere that consists of protons and electrons from the Sun and the Earth's magnetosphere. These charged particles are a known source of ionisation in the polar atmosphere, where the ionisation leads to production of odd nitrogen ($NO_x$) and odd hydrogen ($HO_x$)[1,2]. Both $NO_x$ and $HO_x$ influence ozone balance through catalytic loss cycles[3]. A number of model simulations[4–7] and meteorological reanalysis studies[8–12] have suggested that there could be further implications to the dynamical state of the atmosphere. Changes in atmospheric circulation have been reported all the way to surface level, influencing regional climate variability and seasonal weather conditions[4,5,13]. As we strive towards improved seasonal and climate predictions on regional scales[14–17], we need a better understanding of all sources of natural variability on both annual and decadal scales[18,19]. As part of this, energetic particle forcing is now included as a recommended input for chemistry-climate simulations[20] accounting for solar activity. EPP influences were, for the first time, captured in the Coupled Model

Intercomparison Project Phase 6 (CMIP6) exercise, informing the Intergovernmental Panel on Climate Change Sixth Assessment Report. Thus it is critical that we understand what the implications of solar activity via EPP are on the atmosphere and climate system. The big remaining open question is: What links the well understood upper atmospheric chemical changes from EPP to regional scale dynamics and climate in the troposphere on solar cycle timescales. The main candidate thus far has been the EPP "indirect effect": During the polar winter, EPP-produced $NO_x$ is transported inside the polar vortex from higher altitudes down to the stratosphere[21–23] where ozone loss is initiated[24,25]. However, the transport from lower-thermospheric and mesospheric altitudes, where EPP ionisation is most common, down to the stratosphere takes several months[22]. As a result, the main ozone loss by the EPP-indirect effect takes place during polar spring. This contradicts tropospheric temperature analyses showing changes starting during the winter season[4,13]. Furthermore, springtime stratospheric temperature responses that are highly correlated with

[1]Department of Physics, University of Otago, Dunedin, New Zealand. [2]Space and Earth Observation Centre, Finnish Meteorological Institute, Helsinki, Finland. [3]Sodankylä Geophysical Observatory, University of Oulu, Sodankylä, Finland. [4]Climate and Global Dynamics, National Center for Atmospheric Research, Boulder, CO, USA. [5]Priestley International Centre for Climate, University of Leeds, Leeds, UK. [6]Finnish Meteorological Institute, Helsinki, Finland. ✉e-mail: annika.seppala@otago.ac.nz

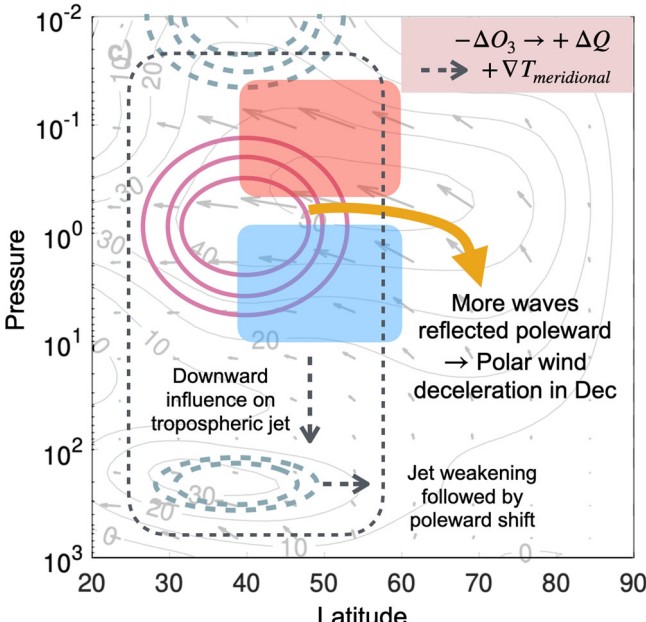

**Fig. 1 | Downward pathways from mesospheric ozone depletion.** Key processes investigated here using controlled model simulations. Zonal mean zonal wind and Eliassen-Palm-flux (EP-flux) from the reference simulation for November are shown in the background. Changes to zonal mean zonal winds are indicated by solid/dashed contours (positive/negative changes) and changes to temperatures indicated by red/blue shading (positive/negative changes). The grey dashed lines indicated a consequence of the shown changes for downward propagation of the signal. Initially, the polar mesospheric ozone depletion results in enhancement of the meridional temperature gradient which drives reduction of the vertical wind shear at mid-latitudes leading to mid-latitude temperature response.

wintertime EPP activity levels have been found to be inconsistent with those expected from in situ ozone changes[8], and careful assessment is required to account for dynamical variability[3]. Considering these pieces of evidence together we hypothesize that early winter chemical-dynamical coupling starts in the mesosphere and plays a major role in transferring any EPP signals downwards.

In this work we test our hypothesis by utilising model simulations where atmospheric chemistry variation along with external forcings are specified, while allowing model dynamics to respond to a controlled change in Arctic polar mesospheric ozone corresponding to observed in situ ozone changes driven by EPP on solar cycle timescales[26,27]. We then use the model simulations to establish the pathway to regional scale variability in the troposphere, particularly focusing on the role of the Quasi Biennial Oscillation (QBO) and effects on the tropospheric jets, and changes to the Northern Annular Mode (NAM) index.

## Results

### Mesospheric ozone loss signal in dynamical variables

While many earlier simulation studies have focused on the indirect stratospheric EPP-$NO_x$ effect, there is now ample evidence of a significant direct impact on mesospheric ozone levels from EPP[27,28]. Satellite observations have proven that EPP events consistently lead to significant production of $HO_x$ and $HO_x$-driven loss of ozone in the mesosphere[27,29], with >30 % ozone variation on solar cycle time scales[6]. Unlike the stratospheric ozone losses discussed above, this does not depend on atmospheric transport processes, but rather takes place in situ at the altitudes where EPP impacts the atmosphere. To investigate the role of this EPP driven mesospheric ozone loss as part of the potential mechanism for driving regional dynamical and climate variability, we applied a corresponding 30% ozone reduction to the

Northern Hemisphere (NH) polar winter mesosphere (~70–80 km) in model simulations (see Methods). To isolate a potential mechanism for the early winter dynamical signals, we focus our experiment solely on the role of the observed mesospheric ozone changes[27], leaving out the stratospheric EPP-indirect effect. A diagram illustrating the dynamical processes under investigation here are presented in Fig. 1, showing key events in early winter.

During the polar night, ozone is an effective emitter of longwave radiation, leading to cooling of the atmosphere. The reduction of ozone in our simulation results in about 0.5 K day$^{-1}$ heating relative to non-perturbed conditions. As seen in the mean temperature fields in Fig. 2 panels a and b, the polar winter mesosphere is already warmer than its' equatorial counterpart due to pole-to-pole circulation, this results in enhancement of the equator-to-pole temperature gradient. On monthly scales (panels e and f), there is further warming of the mid- to high latitude mesosphere, and cooling of the mid-latitude stratosphere in November. Later, in December, the polar mesosphere cools while the polar stratosphere warms by up to 1.25 K. While these changes are initiated by the chemical change, they are not a direct result of it, but rather arise from the dynamical response. The meridional temperature gradient in the atmosphere is balanced by vertical shear of the zonal wind according to the thermal wind balance: an enhancement in the temperature gradient corresponds to a reduction in vertical wind shear. We find evidence for this in our simulations with Fig. 2g showing how the zonal wind at mid-latitudes in November is enhanced below the ozone loss altitudes and reduced above, decreasing the vertical wind shear. This tilts the upper part of the polar vortex towards the equator in November, enhancing the winds in equatorward side of the vortex throughout the stratosphere and lower mesosphere. Wind patterns determine atmospheric wave propagation conditions and changes in wave propagation have been shown to be a key in transferring solar irradiation related dynamical signals via the top-down mechanism[30]. Here, the strengthened mid-latitude winds change the wave guide, resulting in reduction of waves travelling towards the equator with more waves reflected polewards. By December (Fig. 2h) this results in a significant deceleration of polar winds. At the same time the zonal mean tropospheric jet, typically centered below the 100 hPa level around 30°N (panels c and d), moves poleward towards 40°N. This poleward movement of the jet in early winter, explored further in the following sections, agrees with previous studies using reanalysis data[11]. Overall, the weaker polar winds in December would be expected to enable enhanced meridional circulation, resulting in dynamical warming of the polar stratosphere and cooling of the polar mesosphere. This is evident in Fig. 2f which shows clear warming and cooling signals in the polar atmosphere.

### Role of the Quasi Biennial Oscillation

The QBO is known to play a significant role in meridional circulation. It influences the polar atmosphere via the Holton-Tan relationship, resulting in modulation of the polar vortex[31]. During the easterly phase of the QBO (eQBO), the meridional circulation from equator to the pole is intensified, resulting in a weaker polar vortex. The westerly phase of the QBO typically results in opposite conditions. In the context of the solar influence via EPP, a number of studies have suggested links to the QBO[8,11,12,32,33], with indications that the QBO may influence the dynamical coupling from EPP by setting favourable conditions for wave-mean flow interactions, and may itself be influenced by solar activity[32]. The QBO is also known to influence the tropospheric jets via the so-called Subtropical route[31].

Filtering the simulations by the phase of the QBO reveals distinct differences in the response of the atmosphere to the polar mesospheric ozone loss. As seen in Fig. 3 panels c and d, the initial reduction in vertical wind shear at mid-latitudes in November turns into overall weaker polar winds and enhanced equatorward wave propagation during December under eQBO conditions. This enhances meridional

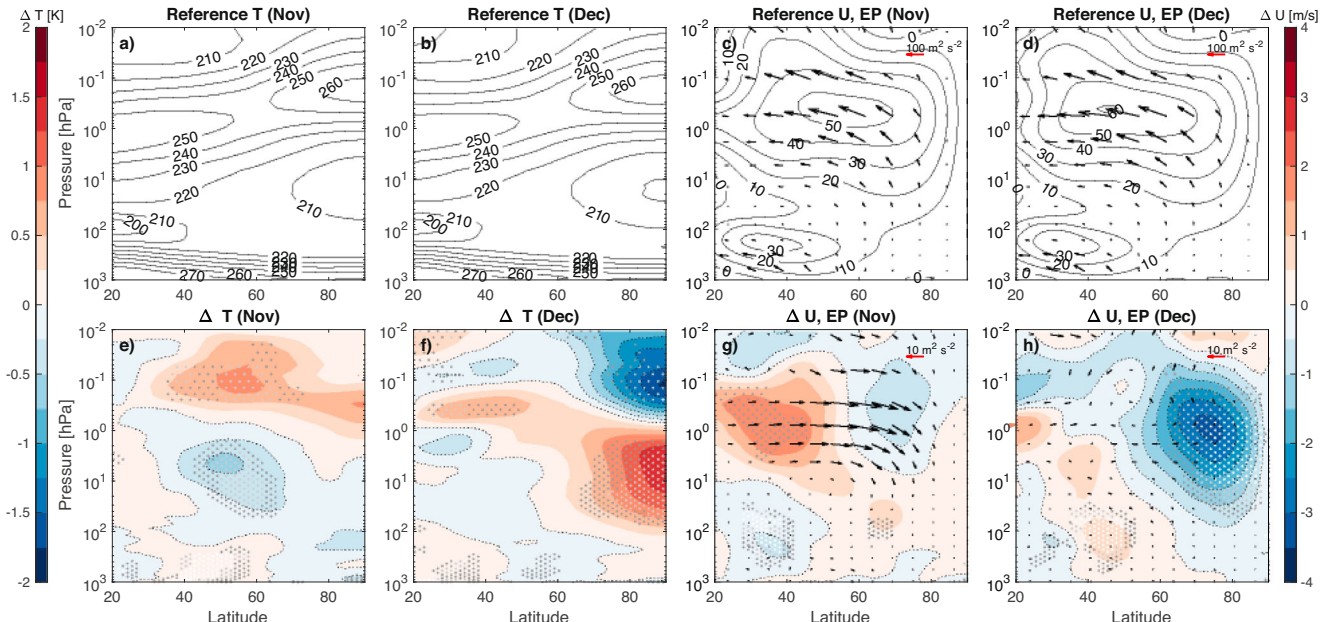

**Fig. 2 | Monthly mean reference fields and composite mean differences.** **a**–**d** November and December mean temperature ($T$, K) and zonal mean zonal wind ($U$, ms$^{-1}$) with overlaid EP-flux (arrows, m²s$^{-2}$, reference scale shown with a red arrow in top right-hand corner) from the reference ("REF") simulations. **e**–**h** Composite mean differences ("EPPO3" simulation − "REF" simulation) for $T$ and $U$ with overlaid EP-flux arrows. Contour intervals are 0.25 K (colourbar on the left) and 0.5 ms$^{-1}$ (colourbar on the right) and for $T$ and $U$, respectively. The reference scale for the EP-flux differences is shown with a red arrow in top right-hand corner. White/light grey/dark grey dots indicate statistical significance of the $T$ and $U$ differences at 95/90/80% levels.

circulation, resulting in heating of the polar stratosphere (and cooling of the polar mesosphere) by up to +2.5 K in December (panel b). Thus the change in polar ozone leads to enhancement of the typical eQBO polar effect in the stratosphere.

During wQBO conditions presented in Fig. 3 panels e–h, the initial change in zonal winds from the mesospheric ozone loss couples to the upper stratospheric QBO wind anomaly, extending this to midlatitudes. This enhances the barrier for equatorward wave propagation in the upper stratosphere, resulting in reduced meridional circulation. By December, the polar vortex edge has shifted equatorwards leading to further reduction in equatorward wave propagation above 10 hPa. Lower down however, waves are now able to propagate from the source region around 50–60°N across the lower stratosphere. Together, these result in an eQBO-like temperature anomaly at mid- and high latitudes[34,35] in December (panel f) with heat redistribution across the stratosphere and lower mesosphere.

While the stratospheric responses in $T$, $U$ and wave forcing in December are very different, in the troposphere both QBO phases show a consistent poleward shift of the tropospheric jet in December, consistent with Fig. 2h. Similar response has previously been reported from reanalysis studies[11,36], and we can now confirm that this takes place as a response to early winter polar mesospheric ozone loss, rather than the EPP-indirect effect. We propose that this is a result of the change in vertical wind shear above the subtropical jet, in effect emulating the QBO Subtropical route[31], but now initiated by the dynamical response to the polar ozone loss.

## Influence on the troposphere

Changes in the tropospheric jet described above are an indication of stratosphere-troposphere coupling[15,37]. To verify any tropospheric implications arising from the polar winter mesospheric ozone loss, we calculated the NAM index in December at the tropospheric pressure level of 500 hPa. The NAM is a dominant mode of dynamical variability in the Northern Hemisphere that is intrinsically linked to variations in the tropospheric jet[15]. It is also frequently used as a climate diagnostic[37].

The distribution of the NAM index from the simulations is presented as a histogram along with the composite differences of the 850 hPa zonal wind showing the eddy-driven tropospheric jet, and the surface level (1000 hPa) temperature in Fig. 4. Without any QBO filtering, the peak of the NAM distribution shifts towards a more positive NAM as a result of the polar mesospheric ozone loss (panel a). This is consistent with the poleward shift of the tropospheric jet[38] seen in Fig. 2h, and we now see in Fig. 4d that this also manifests as strengthening of the poleward edge of the eddy-driven jet located in the Atlantic sector.

While the physical mechanisms have remained unverified, teleconnections between the equatorial QBO and NAM-like patterns in the Northern Hemisphere appear to be real[39]. When filtering the simulations by the QBO phase, in the REF simulation eQBO cases (Fig. 4b) have a clear tendency towards negative NAM (signifying weaker polar vortex during eQBO), while the NAM index distribution is more evenly spread for wQBO (Fig. 4c). In the EPPO3 simulation under eQBO conditions, there is a significant shift (at 90% level) in the peak of the distribution towards a positive NAM, which is consistent with previously found relation with geomagnetic activity (driver of EPP) and early winter NAM[13]. While there is no strengthening of the stratospheric polar vortex during eQBO, the positive NAM signal reflects the poleward shift of the tropospheric jet. For wQBO in panel d, shift towards a positive NAM is not significant; however, the poleward edge of the eddy-driven jet is strengthened (panel f). As in the unfiltered case, the central jet structure appears disrupted, with a tilt or a split. The corresponding surface level temperature differences present a positive NAM like pattern: The significant warming of up to about 1.5 K from Scandinavia to West Siberia in all cases (panels g–i) is similar to those reported previously from seasonal scale analysis[4,7,13]. During eQBO (panel h) there is additional significant cooling (~−1 K) over Europe and Canada and warming (up to 2 K) over parts of United States, which is not present during the wQBO years (panel i). For wQBO, however, the high latitude Arctic in the Pacific sector cools by up to about 1 K. These patterns are broadly consistent with the significant patterns found in reanalysis temperatures in previous studies[13,33].

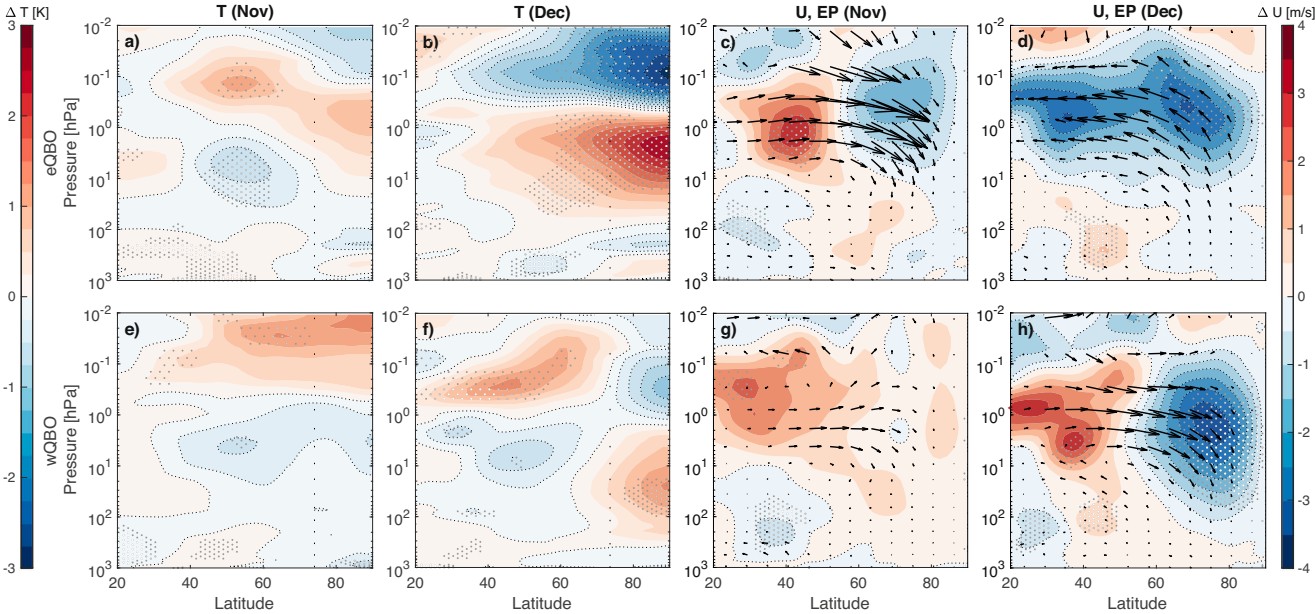

**Fig. 3 | Monthly mean composite differences separated by Quasi Biennial Oscillation (QBO) phase.** As the composite differences in Fig. 2 but now filtered for QBO phase. **a–d** Easterly QBO (eQBO) years only. **e–h** Westerly QBO (wQBO) years only.

## Discussion

Our simulations now show conclusively that the EPP driven mesospheric ozone loss influences the atmosphere below in time scales that are consistent with observations of the tropospheric temperature response[13], solving this long standing problem. We find that this is not solely linked to a stronger polar vortex as has been suggested previously. Rather, we propose that a subtropical route initiated by the changing wind shear above the subtropical jet, similar to the subtropical QBO route, plays a key role for the teleconnection from high latitude mesosphere to the troposphere, finally manifesting as the strengthening of the eddy-driven jet located around 850 hPa.

The positive NAM anomalies in our results provide further evidence of early winter top-down vertical coupling. Wintertime NAM anomalies have previously been linked to EPP forcing[5,9,13], but until now the driver has not been understood. This is highly relevant for the tropospheric climate response as the observed tropospheric temperature anomaly in high EPP conditions[13] corresponds to the typical temperature anomaly under strong polar vortex and positive NAM conditions.

An increasing number of chemistry-climate models are including processes of the stratosphere and mesosphere that are modulated by solar activity. The CMIP6 climate simulations included, for the first time, EPP as part of the recommended solar forcing in past and future climate simulations[20] and this has been retained into CMIP7[40]. While a large motivation behind this has been a more realistic representation of mesospheric and stratospheric ozone variability[41], other benefits from implementation of EPP as part of solar forcing have not been comprehensively assessed. Here we have shown that inclusion of the mesospheric EPP-ozone link has implications for the dynamical conditions in the Northern Hemisphere across a wide range of latitudes and altitudes and is not only limited to the high latitude polar atmosphere. Inclusion of these effects from solar forcing to the eddy-driven jet in model projections will help capture some of the regional climate variability that is currently poorly represented[17,42], providing a currently untapped source of seasonal prediction skill.

While this work is focused on the Northern Hemisphere, similar mechanisms may be taking place in the Southern Hemisphere. Further work, specifically focused on dynamical processes and dominant climate modes in the Southern Hemisphere, is needed to comprehensive

assess the role of EPP driven Antarctic ozone loss on dynamical variability in the Southern Hemisphere. Further consideration should also be given to longer term dynamical implications of EPP driven ozone loss: over longer periods of time, extending into later winter and spring, the EPP-indirect effect will become active and must be taken into account when assessing the pathways to dynamical variability arising from solar activity.

## Methods
### Model
Model simulations were performed using the Specified Chemistry Whole Atmosphere Community Climate Model version 4 (SC-WACCM)[43], a modified version of the CESM/WACCM model. The model has a horizontal resolution of 1.9° latitude by 2.5° longitude and 66 vertical levels, with the model top at $5.1 \times 10^{-6}$ hPa. Simulations presented here use F_2000_WACCM_SC (FWSC) component set of CESM version 1.2. The FWSC compset includes interactive atmosphere and land components. The ocean parameters (e.g. sea surface temperature) and concentrations of radiatively active atmospheric constituents are prescribed based on a prior, fully interactive, integration of the model. Prescribed components corresponds to perpetual year 2000 CE conditions, representing present day conditions. The use of prescribed chemistry allows the control of ozone levels in the simulations. QBO in the model is imposed by relaxing equatorial zonal winds between 86 and 4 hPa to observed interannual variability[44], thus each QBO sample represents the same model years in individual simulations.

### Experiment
To investigate the dynamical processes associated with EPP driven ozone loss in the upper mesosphere (as reported over solar cycle timescales[27]), two sets of simulations were performed for 99 model years each: A modified simulation, "EPPO3", where the Northern Hemisphere polar (60°N–90°N) mesospheric (0.01–0.05 hPa) ozone levels were reduced by 30% from the SC-WACCM background levels during each Northern winter month (November-February). This corresponds to reported observations[27] of EPP impact on polar ozone over roughly monthly timescales. The ozone depletion results in a 0.5 K day$^{-1}$ increase in long wave heating in the EPPO3 simulation, consistent with other experiments[45]. Ozone loss at these pressure

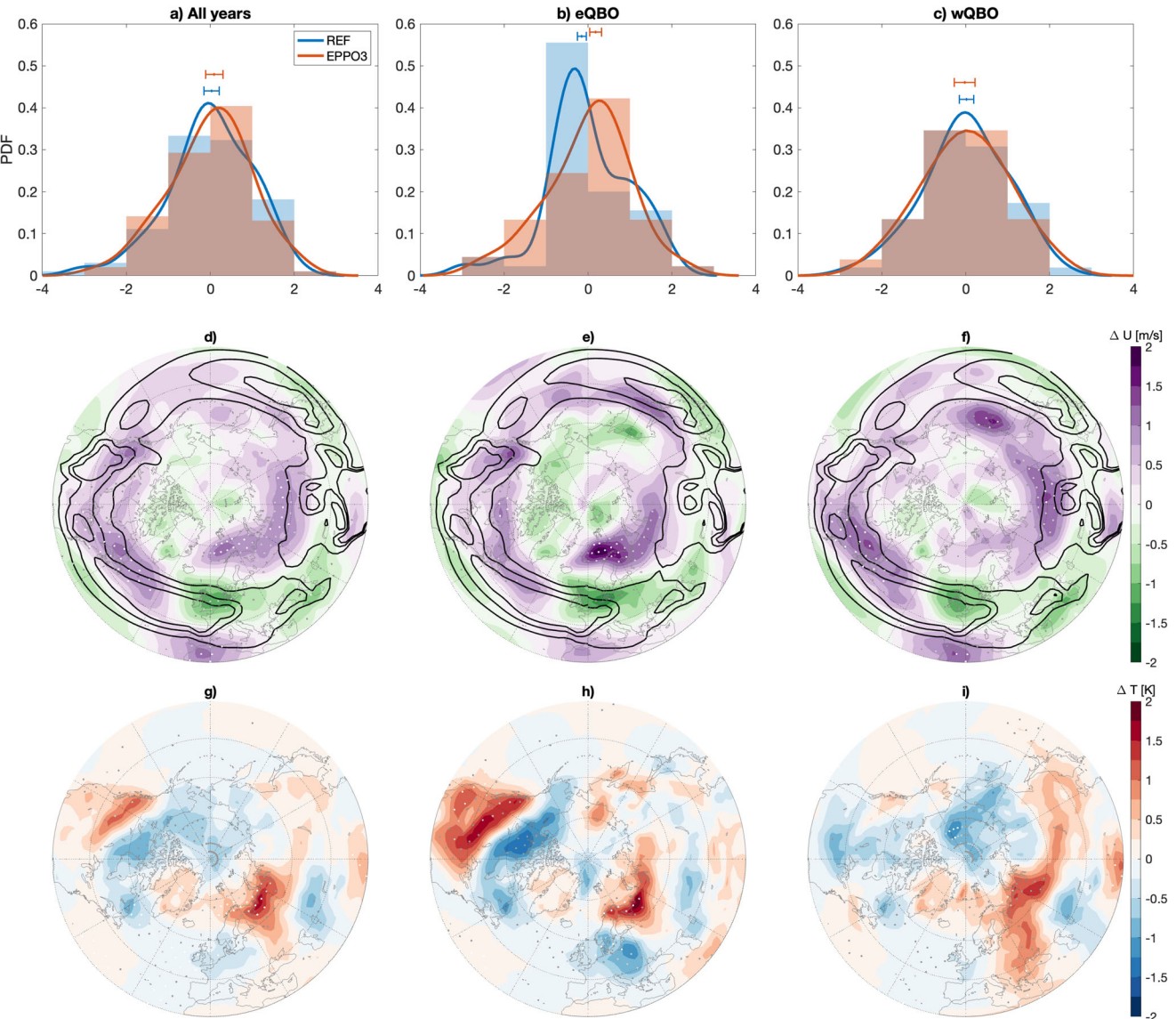

**Fig. 4 | December monthly mean tropospheric anomalies. a–c** December probability density function (PDF) of the Northern Annular Mode (NAM) index at 500 hPa for "EPPO3" (red) and "REF" (blue) simulations: (**a**) All years, (**b**) eQBO only, (**c**) wQBO only. The histograms show the normalised distribution of the NAM index with one increment bins, with lines showing smoothed probability density estimates. Uncertainty estimate for the median NAM index for each case is given at the top, with error bars indicating 90% ranges of the mean of 100,000 bootstrapped medians. **d–f** Zonal wind composite differences "EPPO3" − "REF" ($\Delta U$, ms$^{-1}$) at 850 hPa level for the same cases as in (**a–c**). The black lines show the 5, 8, and 10 ms$^{-1}$ zonal wind contour levels at 850 hPa from the "REF" simulation, to indicate the typical location of the eddy-driven tropospheric jet. **g–i** The corresponding surface level temperature composite differences ($\Delta T$, K). Contour intervals for $U$ (T) are 0.25 ms$^{-1}$ (K) and statistical significance is indicated as before.

levels would correspond to energetic electron precipitation with energies in the range of few hundred keV, or proton precipitation few MeV to about 10 MeV energies[20]. For comparison, the second simulation set, "REF", was performed identically to EPPO3, but without the ozone modification. As we are solely focused on investigating the dynamical implications of controlled ozone depletion, these model simulations enable us to account for internal variability and isolate the effects arising from the polar mesospheric ozone loss. We note that larger ozone depletion has been reported over shorter timescales[27], but our focus here is solely to investigate processes relating to dynamical signals that have been reported at monthly to seasonal scales (relating to solar cycle timescales in EPP variability).

## Analysis
Phase of the QBO was determined from the model equatorial zonal mean zonal wind anomaly at 40 hPa level (Number of years in each

phase for Nov/Dec: #eQBO: 48/47, #wQBO: 51/52). No lag was applied as the investigation is focused on influences of concurrent dynamical conditions.

Statistical significance testing of the results was done using bootstrap re-sampling of the monthly mean fields with 100,000 repetitions[46].

The NAM index for the 500 hPa level is calculated from monthly means of the simulated geopotential height according to the height-dependent empirical orthogonal function (EOF) method[37]. NAM is identified as the leading EOF mode and the NAM index is provided by the principal component of this leading mode. Smoothed NAM probability density distribution estimates were computed using the kernel smoothing estimator (ksdensity) in Matlab. To assess the significance of the NAM peak shifts, we used bootstrap re-sampling with 100,000 repetitions to find a distribution of the median NAM index values in each case assessed. Following the law of large numbers/Central limit

theorem[46], a mean of these bootstrapped medians with a 90% confidence interval based on the distribution of the median values is included with the NAM distributions.

## Data availability
The processed data presented in Figs. 2–4 are available at https://doi.org/10.5281/zenodo.14375389[47].

## Code availability
CESM source code is distributed through a public subversion code repository (http://www.cesm.ucar.edu/models/cesm1.0, last access: 1 November 2024, UCAR, 2024).

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

## Acknowledgements

The work has been supported by the Academy of Finland/Research Council of Finland grant no. 335555 (authors N.K., P.T.V., M.E.S.).

## Author contributions

A.S., P.T.V. and N.K. planned the experiment. N.K. performed the model simulations with support from D.R.M. and M.E.S. A.Y.K. provided expertise in stratosphere-tropospheric coupling. A.S. and N.K. analysed the data and wrote the paper with comments from all co-authors.

## Competing interests

The authors declare no competing interests.
