## [Transparent Peer Review file · Nature Communications]

Polar mesospheric ozone loss initiates downward coupling of solar signal in the Northern Hemisphere

Corresponding Author: Dr Annika Seppälä

Version 0:

Reviewer comments:

Reviewer #1

(Remarks to the Author)

The manuscript suggests that the loss of mesospheric ozone caused by the November-December EEP will lead to a change in tropospheric temperature in December, and QBO with NAM plays an important role in this relationship. Now the results presented in the monograph do not provide convincing conclusions about the discovery of a new mechanism that will show the connection between the destruction of mesospheric ozone and instantaneous changes in surface temperature due to the role of the QBO with NAM.

First, the authors mislead the reader by talking about ozone destruction during the precipitation of energetic particles (EPP), although estimates of ozone destruction during the precipitation of electrons from radiation belts (EEP) are used for modeling. If authors model the effects of electron precipitation, then it is necessary to adjust the title and text of the manuscript. In any case, the words "solar signal" in the title are too abstract to describe the actual results. If the authors want to demonstrate the effect of ozone destruction from the deposition of energetic particles of protons and electrons (EPP), then it is necessary to correctly insert references and clearly determine the percentage of ozone destruction from the precipitation of both electrons and protons. Introduce readers to the concepts of EEP and EPP.

Secondly, the authors use previously made estimates of ozone depletion by 30% during the solar cycle for their model calculations and estimates during the first two winter months. It is physically incorrect to assume that a 30% percent loss of ozone from EEP will occur every month for decades to come. By what percentage should ozone be destroyed in November for the mechanism presented in this manuscript to work?

Thirdly, it is not at all clear how long (hours to days) ozone destruction must occur since the beginning of November for the hypothetical QBO-related mechanism to work and effects in surface temperatures to be visible? Needs confirmation from observations that show that the destruction of mesospheric polar ozone in November by at least 5-30% will lead to variations in surface temperatures in December by about 1K in different regions of the planet.

Some points explaining the statements presented above:

Page 2, line 23: Reference [5] presents results for EEP (Energetic Electron Precipitation) over a solar cycle (11 years) rather than for winter months. The paper [5] does not provide results for EPP (Energetic Particle Precipitation: protons and electrons). The ozone loss varies from 5 till 30% based on GOMOS, SABER, MLS during about 12 days. In model calculations, ozone destruction was assumed to be 30% from November to February, then in November it turns out that ozone was destroyed by 7.5% during about 10 days and such ozone destruction leads to changes in surface temperatures by about 1 K already in December. This is true?

Page 3, line 41: The knowledge that the production of NO_x and HO_x is caused by the ionization rate is very well known even before Veronnen et al 2013. The first original works should be included here as references and more references should be provided.

Page 3, lines 50-52: Here should be added references confirming that EPP CMIP6 lead to some conclusion in IPCC Sixth Assessment Report.

Page 4, lines 70-72: The manuscript contains a large number of self-citations of results that often do not reflect the real situation. For example, there are many scientific articles and review papers that show ozone loss under SPE, EEP or EPP. Reference [26] is a self-cited work discusses the increase in nitric acid concentrations in the mesosphere during solar proton events. Reference [5] does not examine ozone depletion due to EPP at all. Authors should exclude numerous self-citations of works and use links to those articles that truly reflect the processes being discussed.

Page 4 Methods and pages 12-13 Methods: It is written that two 99-year model experiments were conducted, one as a reference and the other with ozone depletion set to 30% in November-February. This explanation is not sufficient and it is necessary to clarify how ozone depletion was prescribed - 30% or 7.5% per year for each month for 99 years, or 30% per solar cycle, as described in reference [5]. How many days in a month does ozone depletion last?

There is no explanation why the period of 99 years was taken. How will the results change if, for example, take 11 model years as solar cycle and conduct, 5 model experiments? Currently, there is no level of confidence in the model results. Can the authors replicate the results with the nudge for selected years and compare them with observations? These results should confirm the mechanism of EEP signal propagation due to the loss of ozone from the mesosphere to the troposphere under the influence of QBO and NAM over several weeks. Otherwise, the results described here are not obvious and are based only on synthetic assumptions that are far way from the real.

Reviewer #2

(Remarks to the Author)

The article "Polar mesospheric ozone loss initiates downward coupling of solar signal" deals with the relation of energetic particle precipitation (EPP) with solar activity and climate variability. The paper present results from simulations where polar winter upper mesospheric ozone is reduced to levels corresponding to observed in situ ozone changes driven by EPP. This work finds that the large scale background atmospheric conditions determined by the Quasi Biennial Oscillation (QBO) influence the stratospheric response to EPP, but significant responses in the tropospheric eddy-driven jet take place broadly consistently for both QBO phases. The authors provide compelling numerical evidence that the signals from the initial EPP driven in situ ozone changes at high mesospheric altitudes propagate downwards in timescales that correspond to observed tropospheric level climatic changes linked to EPP. The paper provides a key new way to understanding stratosphere-troposphere coupling.

In my opinion, the scientific results are novel and the methodologies are well described. I only have two minor comments after which the paper might be suitable for publication:

1. What energy ranges of particles are relevant here? Usually energetic particles refer to particles with energies several time higher than the thermal speed. Have the authors done any testing with varying energy level?
2. In the "Model" section it is specified that "The model has a horizontal resolution of 1.9° latitude by $214 \times 2.5^\circ$ longitude and 66 vertical levels, with the model top at 5.1×10^{-6} hPa". Have the authors checked whether changing these simulation parameters affect the results drastically?

Reviewer #3

(Remarks to the Author)

Review NCOMMS-24-18123-T, 01Aug2024

Key results

The authors modeled the influence on lower atmosphere layers of the EPP during northern winter, which leads to a decrease in the ozone content in the polar mesosphere. The loss of ozone leads to a decrease in the cooling of the atmosphere, which changes the dynamics of the circulation, moving this signal down to the stratosphere and upper troposphere and shifting the tropospheric jet poleward. The authors propose that these processes can significantly affect the regional climate down to the middle latitudes. Their new mechanism makes it possible to trace the influence of EPP events on the troposphere outside the polar latitudes, in contrast to the known mechanism of EPP-indirect influence. The modeling was carried out using SC-WACCM V4. In addition, the authors showed a change in the degree of the EPP influence on the dynamics of the atmosphere depending on the QBO phase. The article is of considerable interest and is in scope of the journal, however, it needs further clarification of several issues (see below).

Validity

The modeling results (shown in Figures 1–3) appear to be reasonable and correctly interpreted. In the authors' opinion, the results should allow us to improve our understanding of the mesosphere-stratosphere-troposphere coupling processes. However, are the given illustrations sufficient to reliably interpret the processes that occur when "the EPP source is switched on" for four winter months and assuming that there is a 30% ozone deficiency in the polar mesosphere of the Northern Hemisphere during these four months? At the same time, the authors consider only two months, November and December, and draw conclusions about the time scale of processes with a discreteness of one month. It would be much more convincing if the modeling results (reduced ozone in November–February and reference calculations) were obtained for a time interval of at least half a year: from September to April. Maybe it is not necessary to provide these results in the manuscript, but to explain in the text what happens over a longer period of time.

Significance

The results obtained show that signals from EPP-induced mesospheric ozone changes can propagate downwards to lower latitudes on observational timescales and can influence regional climate. The paper provides new consideration into the influence of EPP-induced polar mesospheric ozone changes on tropospheric processes. Due to the chain of processes proposed by the authors, this should significantly extend our understanding of the influence of energetic charged particles on tropospheric parameters.

Data and methodology

The data are obtained by the authors using modeling of a 30% reduction in mesospheric ozone, which is consistent with the observed variations and appears to be reasonable. However, to confirm the obtained results and conclusions, it is necessary either to discuss in more detail the results shown in Fig. 3 or to provide additional modeling results confirming the connections and time scale of the response of the mid-latitude troposphere to EPP events (more precisely, to the long-term decrease in ozone in the mesosphere during the four winter months).

Analytical approach

In my opinion, the idea of the authors' modeling is reliable and shows that the decrease in the signal from the EPP-induced mesospheric ozone can descend into the stratosphere and troposphere and extend to mid-latitudes. However, it is not clear how the authors retrieved a characteristic time frame and confirmed that it is close to the observed rate of tropospheric (temperature) change. Yes, the assumption introduced in the model that the polar mesosphere ozone decreases by 30% during the Northern Hemisphere winter is reasonable. But why did they discuss only one month (December) of this influence? The few months before the mesosphere ozone depletion mechanism is "turned on" and the few months after it is "turned off" would be very useful to consider.

Comments and suggested improvements

1. It would be useful to present a figure or diagram at the beginning of the manuscript illustrating the sequence of processes under consideration.
2. It is not clear from the manuscript whether the proposed consideration and mechanism equally concern the polar latitudes of the Northern and Southern hemispheres. Looking at the Title, someone can think that the manuscript deals with both polar regions. Should the Northern Hemisphere be included in the title of the article?
3. It would be useful to briefly consider the applicability of the proposed mechanism to the Southern Hemisphere atmosphere and to suggest what differences in the processes there may be. Although this may be the subject of another publication, a short respective paragraph in this manuscript would be useful.
4. Figure 3 raises the most questions. Firstly, it is technically logical to sign individual images as in Figures 1 and 2 as a) – i), then it is easier to describe the results shown in them. Then there is no need to describe – "the top row", "in the middle", "on the right", etc.
5. It is difficult to assess the accuracy and significance of the results in the top row of Figure 3 because the accuracies are not indicated.
6. Lines 152–153. How and why NAM is calculated at a pressure level of 500 hPa – it is necessary to define how the NAM index is calculated in this case, why it is related to the tropospheric pressure level of 500 hPa.
7. Figure 3, Middle. "zonal wind contour levels at 850 hPa from the REF simulation, an indication for the locations of the tropospheric jet" – How can the tropospheric jet be located at 850 hPa when the mean tropospheric jet pressure level is about 200 hPa? Should be clarified.
8. The discussion at Lines 169–191 needs better argumentation and clarification. For example, Lines 176–177 "The warming of up to about 1.5 K in the Northern Eurasian sector" – what is "Northern Eurasian sector"? Actually, this warming area covers areas from Scandinavia to West Siberia, but not entire Eurasia. Need to be more specific. Line 179, "and warming (up to 2 K) over North America" – actually, warming is seen over part of US, but not over the whole North America that extended from Canada to Panama. Note that there is a significant cooling (about –1.2 K) over north Canada, which is not discussed.
9. How can the time scale of the EPP influence on troposphere temperature changes be retrieved if authors compare data in two consecutive months, November and December when the source is "switched on" for four months (November–February)? The authors discuss the effect in the second month in a time when the source is still "switched on"? The physical base of the experiment needs to be clarified. What purpose do you need to reduce ozone for four months if you consider just two first months? It would be reasonable to reduce ozone for one month (November) and look for effect next month in December. Or look at the effect in March and April, when the ozone reduction is switched off.
10. It is worth more clearly explaining what physical process is modeled by the permanent mesospheric ozone decrease during four months each year. The characteristic time of short term EPP events is equal to days, and long-term modulation of the mesospheric ozone concentration happens with the solar cycle period near to 11 years.

Clarity and context

The text is written in clear language and does not leave room for misinterpretation. The authors have presented the problem in necessary detail and have provided a sufficient number and quality of peer-reviewed references to previous published works by other authors to clarify the context of the proposed study.

References

In my opinion, the manuscript includes the necessary quantity and quality of previous literature.

Technical comment

Lines 244-246. Two authors do not have any specific function.

Finally,

what are the primary and new results? Is there a change in tropospheric temperature by 1–2K in some regions (bottom panel of Fig. 3)? It should be more explicit and emphasized in the text. Generally, authors proposed a fruitful hypothesis when the early winter (November) process of the polar mesospheric ozone reduction under EPP impact produces the chain of signals propagating from the mesosphere through the stratosphere to the lower tropospheric altitudes in mid-latitudes. However, this hypothesis still needs a more detailed explanation of the processes chain and modeling results that authors can provide. That can help increase the significance of the results obtained, which is really important to better understanding EPP-mesosphere-stratosphere-troposphere coupling processes.

Reviewer #4

(Remarks to the Author)

Version 1:

Reviewer comments:

Reviewer #1

(Remarks to the Author)

The manuscript still needs some improvements.

1) The authors of the manuscript "Polar mesospheric ozone loss initiates downward coupling of solar signal in the Northern Hemisphere" base their theory on tropospheric temperature changes in December, suggesting a dynamic coupling of the mesosphere, stratosphere, and troposphere that began with a 30% reduction in mesospheric ozone due to EPP forcing in November.

The authors strongly favor the term "energetic particle precipitation (EPP)" but cite only one paper that estimates % ozone depletion by "energetic electron precipitation (EEP)". [Andersson, M. E., Verronen, P. T., Rodger, C. J., Clilverd, M. A. & Seppälä, A. Missing driver in the Sun-Earth connection from energetic electron precipitation impacts mesospheric ozone. *Nature Comm.* 5, D07307 (2014).]

If authors want to use the phrase energetic particle precipitation, they should add some references that estimate % ozone depletion by energetic proton precipitation (during solar energetic particle events (SEP)). Many studies have reported a decrease in mesospheric ozone of approximately several tens of percent associated with separate SEPs [e.g. Weeks et al., 1972; McPeters et al., 1981; Thomas et al., 1983; Jackman et al., 2008; etc.] and with SEPs over solar cycles [McPeters and Jackman, 1985; Rozanov et al., 2012; Doronin et al., 2024]

Please add at least the suggested references:

[McPeters, R. D., and C. H. Jackman (1985), The response of ozone to solar proton events during solar cycle 21: The observations, *J. Geophys. Res.*, 90(D5), 7945–7954, doi:10.1029/JD090iD05p07945.]

[Rozanov, E., Calisto, M., Egorova, T. et al. (2012), Influence of the Precipitating Energetic Particles on Atmospheric Chemistry and Climate. *Surv Geophys* 33, 483–501. <https://doi.org/10.1007/s10712-012-9192-0>]

[Doronin, G.; Mironova, I.; Bobrov, N.; Rozanov, E. (2024), Mesospheric Ozone Depletion during 2004–2024 as a Function of Solar Proton Events Intensity. *Atmosphere*, 15, 944. <https://doi.org/10.3390/atmos15080944>]

in the text of the manuscript:

Page 1, 6th line from the bottom

"ozone changes driven by EPP on solar cycle timescales"

Page 3, 2nd line from the bottom

"with >30 % ozone variation on solar cycle time scales"

Page 10, 2nd line from top

"EPP driven ozone loss in the upper mesosphere (as reported over solar-cycle timescales)"

2) It is hard to believe that the mesospheric ozone layer depletes by 30% each November over several solar cycles. However, proving this is not the purpose of this paper.

But it should be explained correctly that the authors are using temperature changes in December based only on the EPP forcing during November.

Include changes in the phrase "by 30% from the SC-WACCM background levels during northern winter months (November-February)", clarifying that the authors mean, "by 30% the SC-WACCM background levels during each northern winter month".

Reviewer #2

(Remarks to the Author)

The authors have now addressed all the comments. I believe the manuscript is acceptable for publication.

Reviewer #3

(Remarks to the Author)

I appreciate the authors' updates and responses. This revision has greatly improved the manuscript. I reviewed everything carefully and can confirm that the authors have significantly improved the text and practically fulfilled all my "requests". I believe that based on this revision, the article can be published. Since I discussed key results, validity, and significance of the manuscript results in detail in my previous review, I think there is no need to repeat the level and rating here. The inclusion of Figure 1 to illustrate the processes being investigated greatly clarifies the scope and the relationships and sequences being investigated. The section on data and methodology, as well as the analytical approach, have been improved and clarified. I recommend that this manuscript be published.

Reviewer #4

(Remarks to the Author)

Author responses to reviewer comments on paper "Polar mesospheric ozone loss initiates downward coupling of solar signal".

We would like to thank all reviewers for their comments. Our detailed responses to all comments are included here.

Comments from Reviewer # 1

General comments:

The manuscript suggests that the loss of mesospheric ozone caused by the November-December EEP will lead to a change in tropospheric temperature in December, and QBO with NAM plays an important role in this relationship. Now the results presented in the monograph do not provide convincing conclusions about the discovery of a new mechanism that will show the connection between the destruction of mesospheric ozone and instantaneous changes in surface temperature due to the role of the QBO with NAM.

First, the authors mislead the reader by talking about ozone destruction during the precipitation of energetic particles (EPP), although estimates of ozone destruction during the precipitation of electrons from radiation belts (EEP) are used for modeling. If authors model the effects of electron precipitation, then it is necessary to adjust the title and text of the manuscript. In any case, the words "solar signal" in the title are too abstract to describe the actual results. If the authors want to demonstrate the effect of ozone destruction from the deposition of energetic particles of protons and electrons (EPP), then it is necessary to correctly insert references and clearly determine the percentage of ozone destruction from the precipitation of both electrons and protons. Introduce readers to the concepts of EEP and EPP.

Secondly, the authors use previously made estimates of ozone depletion by 30% during the solar cycle for their model calculations and estimates during the first two winter months. It is physically incorrect to assume that a 30% percent loss of ozone from EEP will occur every month for decades to come. By what percentage should ozone be destroyed in November for the mechanism presented in this manuscript to work?

Thirdly, it is not at all clear how long (hours to days) ozone destruction must occur since the beginning of November for the hypothetical QBO-related mechanism to work and effects in surface temperatures to be visible? Needs confirmation from observations that show that the destruction of mesospheric polar ozone in November by at least 5-30% will lead to variations in surface temperatures in December by about 1K in different regions of the planet.

Comment: Page 2, line 23: Reference [5] presents results for EEP (Energetic Electron Precipitation) over a solar cycle (11 years) rather than for winter months. The paper [5] does not provide results for EPP (Energetic Particle Precipitation: protons and electrons). The ozone loss varies from 5 till 30% based on GOMOS, SABER, MLS during about 12 days. In model calculations, ozone destruction was assumed to be 30% from November to February, then in November it turns out that ozone was destroyed by 7.5% during about 10 days and such ozone destruction leads to changes in surface temperatures by about 1 K already in December. This is true?

Reply: Energetic Particle Precipitation (EPP) is an umbrella term that encompasses solar proton events, energetic electron precipitation due to auroral processes as well as electrons from the radiation belts, and cosmic rays. The use of EPP as the overall term has a settled meaning in both the scientific community and main stream literature. In particular, it is used for the scientific community publications for the CMIP (Climate Model Intercomparison Project) activities. Importantly here, all above result in enhanced ionisation in the atmosphere. The ionisation results in the same outcomes (i.e. ozone loss), thus for the atmospheric impact, the original source doesn't matter. What matters is the enhance ionisation resulting in ozone loss. At a high level journal such as Nature Communications we should use accurate terms without field specific jargon, thus we feel strongly that the general term EPP is the right to use here.

Since this study is solely focused on the theory explaining the potential dynamical mechanisms that could operate in a timescale that matches observations, we have applied an idealised ozone loss to control for any other variability and isolate the dynamical mechanism. This is at 30% level for each month as explained in the methods section. We do not understand where the reviewer is taking 7.5% from (percentages from different months, mathematically, can not be added up). If each month would have had a 7.5% ozone depletion as the reviewer seems to think, this would still only add to 7.5%.

Comment: Page 3, line 41: The knowledge that the production of NO_x and HO_x is caused by the ionization rate is very well known even before Veronnen et al 2013. The first original works should be included here as references and more references should be provided.

Reply: Verronen et al. (2013) provided a comprehensive chemical background on this topic, however, as requested we have replaced this reference with the older Rusch et al. (1981) and Solomon et al. (1981).

Comment: Page 3, lines 50-52: Here should be added references confirming that EPP CMIP6 lead to some conclusion in IPCC Sixth Assessment Report.

Reply: We are not sure what exactly the reviewer means by these references as we are not aware of EPP specific conclusions (as discussed in our Discussion section). We have cited Matthes et al. (2017) for description of the EPP forcing as well as the atmospheric impacts of it. The IPCC report is publicly available, we are not in a position to summarise the reports findings in the present research manuscript. Note that we also now cite the CMIP7 recommendation in our manuscript, as CMIP7 again includes EPP forcing.

Comment: Page 4, lines 70-72: The manuscript contains a large number of self-citations of results that often do not reflect the real situation. For example, there are many scientific articles and review papers that show ozone loss under SPE, EEP or EPP. Reference [26] is a self-cited work discusses the increase in nitric acid concentrations in the mesosphere during solar proton events. Reference [5] does not examine ozone depletion due to EPP at all. Authors should exclude numerous self-citations of works and use links to those articles that truly reflect the processes being discussed.

Reply: A number of the authors (Seppälä, Verronen, Marsh, Szeglag) have each published several key scientific discoveries in the field of EPP impacts on the atmosphere. As such, the citations reflect that. We would like to highlight that we cite a number of other authors: e.g. Rozanov, Randall, Arsenovic, Lu, Li, Salminen, Funke, Damiani, Maliniemi and Meraner, all presenting work highly relevant to the processes discussed in the present manuscript. Over 70% of the reference cited do not include the authors of the manuscript as lead, or co-authors. For example Reference [5], titled *Missing driver in the Sun-Earth connection from energetic electron precipitation impacts mesospheric ozone*, is entirely a about EPP impact on ozone (two of the three figures in the article specifically present satellite observations of ozone depletion after EPP).

Comment: Page 4 Methods and pages 12-13 Methods: It is written that two 99-year model experiments were conducted, one as a reference and the other with ozone depletion set to 30% in November-February. This explanation is not sufficient and it is necessary to clarify how ozone depletion was prescribed - 30% or 7.5% per year for each month for 99 years, or 30% per solar cycle, as described in reference [5]. How many days in a month does ozone depletion last? There is no explanation why the period of 99 years was taken. How will the results change if, for example, take 11 model years as solar cycle and conduct, 5 model experiments? Currently, there is no level of confidence in the model results. Can the authors replicate the results with the nudge

for selected years and compare them with observations? These results should confirm the mechanism of EEP signal propagation due to the loss of ozone from the mesosphere to the troposphere under the influence of QBO and NAM over several weeks. Otherwise, the results described here are not obvious and are based only on synthetic assumptions that are far way from the real.

Reply: We are not sure why the reviewer thinks 7.5% depletion was applied, we do not suggest this anywhere. If ozone was reduced in each month from November to February by 7.5%, this would only add up to 7.5% if integrated over time. This could never mathematically add up to 30% and is not what we have done. As we explain in the Methods, we reduce ozone by 30% in each month mentioned, the same level of reduction (30%) is applied throughout the whole time, not only for some days of the month. We have added further clarification to the methods section, emphasizing that ozone is reduced 30% from the fixed model background.

A 99-year experiment using specified chemistry provides us with a large sample of results with identical forcing, but varying initial conditions for each year providing background dynamical variability. 5×11 years, as suggested by the reviewer, would only yield 55 years - a smaller sample. We now have 99 years of no EPP forcing (i.e. quiet period of activity) and 99 years of average active period (2×99 years = 198 years).

We control for all other chemical and external variability apart from the polar ozone loss specifically in order to isolate dynamical mechanisms. Hence there is no solar cycle. As stated in the Methods section, the model external forcings are fixed to the year 2000 and not allowed to vary. Thus these simulations correspond to the average maximal ozone depletion reported in previous observational studies. This is the only possible way to extract the mechanisms from ozone loss to dynamics. Nudging the model, which means forcing the model's dynamics towards a specific state, would make it impossible to determine anything about dynamics – simulations such as these would be meaningless. Thus we are not sure why the reviewer would recommend nudging approach. The goal of using the specified chemistry version of the model is specifically to investigate dynamical response to set chemical conditions.

We would like to emphasise that this study is investigating the dynamical mechanisms that would enable linking mesospheric ozone loss to tropospheric level response. We are not trying to replicate conditions of a specific year as this would not aid in entangling the dynamical mechanisms.

Comments from Reviewer # 2 The article “Polar mesospheric ozone loss initiates downward coupling of solar signal” deals with the relation of energetic particle precipitation (EPP) with solar activity and climate variability. The paper present results from simulations where polar winter upper mesospheric ozone is reduced to levels corresponding to observed in situ ozone changes driven by EPP. This work finds that the large scale background atmospheric conditions determined by the Quasi Biennial Oscillation (QBO) influence the stratospheric response to EPP, but significant responses in the tropospheric eddy-driven jet take place broadly consistently for both QBO phases. The authors provide compelling numerical evidence that the signals from the initial EPP driven in situ ozone changes at high mesospheric altitudes propagate downwards in timescales that correspond to observed tropospheric level climatic changes linked to EPP. The paper provides a key new way to understanding stratosphere-troposphere coupling.

In my opinion, the scientific results are novel and the methodologies are well described. I only have two minor comments after which the paper might be suitable for publication:

Comment: What energy ranges of particles are relevant here? Usually energetic particles refer to particles with energies several time higher than the thermal speed. Have the authors done any testing with varying energy level?

Reply: For ozone depletion impacts in the upper mesosphere, precipitating electrons would have energies of few hundred keV. These are higher than typical auroral electrons of < 10 keV but could relate to radiation belt or substorm driven electron precipitation. For precipitating protons this would require few MeV to about 10 MeV energies. We now write in the Methods section that *Ozone loss at these pressure levels would correspond to energetic electron precipitation with energies in the range of few hundred keV, or proton precipitation few MeV to about 10 MeV energies.* citing Matthes et al. (2017).

We have not tested different energies here, as our analysis fixes the ozone loss levels to those corresponding to observations over longer periods of time as reported by Andersson et al. (2014).

Comment: In the “Model” section it is specified that “The model has a horizontal resolution of 1.9° latitude by 2.5° longitude and 66 vertical levels, with the model top at 5.1×10^{-6} hPa”. Have the authors checked whether changing these simulation parameters affect the results drastically?

Reply: We have not tested this as we used the available model resolution for this model version, as described in the cited work of Smith, K. L., Neely, R. R., Marsh, D. R. & Polvani, L. M. The Specified Chemistry Whole Atmosphere Community Climate Model (SC-WACCM), *J. Adv. Mod. Earth Sys.* 6, 883–901 (2014). To be able to test other resolutions would unfortunately first require model development and testing work.

Comments from Reviewer # 3

Key results.

The authors modeled the influence on lower atmosphere layers of the EPP during northern winter, which leads to a decrease in the ozone content in the polar mesosphere. The loss of ozone leads to a decrease in the cooling of the atmosphere, which changes the dynamics of the circulation, moving this signal down to the stratosphere and upper troposphere and shifting the tropospheric jet poleward. The authors propose that these processes can significantly affect the regional climate down to the middle latitudes. Their new mechanism makes it possible to trace the influence of EPP events on the troposphere outside the polar latitudes, in contrast to the known mechanism of EPP-indirect influence. The modeling was carried out using SC-WACCM V4. In addition, the authors showed a change in the degree of the EPP influence on the dynamics of the atmosphere depending on the QBO phase. The article is of considerable interest and is in scope of the journal, however, it needs further clarification of several issues (see below).

Validity.

The modeling results (shown in Figures 1–3) appear to be reasonable and correctly interpreted. In the authors' opinion, the results should allow us to improve our understanding of the mesosphere-stratosphere-troposphere coupling processes. However, are the given illustrations sufficient to reliably interpret the processes that occur when "the EPP source is switched on" for four winter months and assuming that there is a 30% ozone deficiency in the polar mesosphere of the Northern Hemisphere during these four months? At the same time, the authors consider only two months, November and December, and draw conclusions about the time scale of processes with a discreteness of one month. It would be much more convincing if the modeling results (reduced ozone in November–February and reference calculations) were obtained for a time interval of at least half a year: from September to April. Maybe it is not necessary to provide these results in the manuscript, but to explain in the text what happens over a longer period of time.

Significance.

The results obtained show that signals from EPP-induced mesospheric ozone changes can propagate downwards to lower latitudes on observational timescales and can influence regional climate. The paper provides new consideration into the influence of EPP-induced polar mesospheric ozone changes on tropospheric processes. Due to the chain of processes proposed by the authors, this should significantly extend our understanding of the influence of energetic charged particles on tropospheric parameters.

Data and methodology.

The data are obtained by the authors using modeling of a 30% reduction in mesospheric ozone, which is consistent with the observed variations and appears to be reasonable. However, to confirm the obtained results and conclusions, it is necessary either to discuss in more detail the results shown in Fig. 3 or to provide additional modeling results confirming the connections and time scale of the response of the mid-latitude troposphere to EPP events (more precisely, to the long-term decrease in ozone in the mesosphere during the four winter months).

Analytical approach.

In my opinion, the idea of the authors' modeling is reliable and shows that the decrease in the signal from the EPP-induced mesospheric ozone can descend into the stratosphere and troposphere and extend to mid-latitudes. However, it is not clear how the authors retrieved a characteristic time frame and confirmed that it is close to the observed rate of tropospheric (temperature) change. Yes, the assumption introduced in the model that the polar mesosphere ozone decreases by 30% during the Northern Hemisphere winter is reasonable. But why did they discuss only one month (December) of this influence? The few months before the mesosphere ozone depletion mechanism is "turned on" and the few months after it is "turned off" would be very useful to consider.

Author reply: We are very grateful for the detailed comments on the manuscript. Perhaps the most comprehensive earlier modelling study that aimed to determine the potential climate implications of

EPP drive ozone loss is that of Meraner and Schmidt (2018, included in our reference list). Their simulation design aimed to assess both the direct impact on mesospheric ozone levels, as well as the EPP-indirect effect. They analyse results on seasonal scale (e.g. DJF) and, while not discussed, there is clearly a large QBO bias present. Due to the transient nature of the QBO progression this could be difficult to control for on seasonal scale analysis. In light of their results and other works suggesting that QBO plays a role in the downwards transport of the EPP signal, we have now been able to more robustly account for the QBO effect by limiting to monthly scale analysis. Extending the analysis into the spring months requires accounting for the EPP-indirect effect, and while this should be a focus of further study, we wanted to mainly tackle the problem of reported early winter tropospheric level changes. We would be happy to include results from the extended winter-spring months as supplementary material if this is seen as necessary, but we are concerned that these are not representative of the full EPP impacts – they do not capture the EPP-indirect effect which would be present following a winter of EPP activity.

Comment: It would be useful to present a figure or diagram at the beginning of the manuscript illustrating the sequence of processes under consideration.

Reply: We have added a diagram (new Figure 1) to illustrate processes under investigation as requested. We focused this diagram on the early winter processes as this provided most clarity.

Comment: It is not clear from the manuscript whether the proposed consideration and mechanism equally concern the polar latitudes of the Northern and Southern hemispheres. Looking at the Title, someone can think that the manuscript deals with both polar regions. Should the Northern Hemisphere be included in the title of the article?

Reply: We have added “Northern Hemisphere” into the title as suggested to emphasize that the work is focused there.

Comment: It would be useful to briefly consider the applicability of the proposed mechanism to the Southern Hemisphere atmosphere and to suggest what differences in the processes there may be. Although this may be the subject of another publication, a short respective paragraph in this manuscript would be useful.

Reply: Much of the work on tropospheric implications has thus far focused on the Northern Hemisphere, particularly on the Northern Annular Mode (NAM). Similar mechanisms maybe in acting in the Southern Hemisphere, but as far as we know there is less published work discussing effects on the dominating tropospheric modes of variability. It is also possible that because the wave-mean flow interaction is generally less affected in the Southern Hemisphere, such chemical changes in the polar mesosphere would not be enough to disturb the already strong polar vortex. We now write in the discussion section of the manuscript: *While this work is focused on the Northern Hemisphere, similar mechanisms may be taking place in the Southern Hemisphere. Further work, specifically focused on dynamical processes and dominant climate modes in the Southern Hemisphere, is needed to comprehensive assess the role of EPP driven Antarctic ozone loss on dynamical variability in the Southern Hemisphere.*

Comment: Figure 3 raises the most questions. Firstly, it is technically logical to sign individual images as in Figures 1 and 2 as a) – i), then it is easier to describe the results shown in them. Then there is no need to describe – “the top row”, “in the middle”, “on the right”, etc.

Reply: We have implemented this for the Figure as suggested and now refer to panels a-i.

Comment: It is difficult to assess the accuracy and significance of the results in the top row of Figure 3 because the accuracies are not indicated.

Reply: We have now assessed the significance of the NAM results by performing bootstrapping with 100,000 repetitions to determine the peak and its uncertainty for each case. We have included this as a 90% significance range, which further clarifies that the eQBO case is where the shift towards positive NAM is significant. This information with error bars has been added to the figure and we have added details of the calculation into the Methods section.

Comment: Lines 152–153. How and why NAM is calculated at a pressure level of 500 hPa – it is necessary to define how the NAM index is calculated in this case, why it is related to the tropospheric pressure level of 500 hPa.

Reply: The NAM calculation method was included in the Methods-section of the manuscript: *The Northern Annular Mode index for the 500 hPa level is calculated from monthly means of the simulated geopotential height according to the height dependent empirical orthogonal function (EOF) method*³⁵. *Smoothed NAM probability density distribution estimates were computed using the kernel smoothing estimator (ksdensity) in Matlab.* As stated we follow the calculation method presented by Baldwin and Thompson (2009) (original reference 35.), for the height-dependent EOF method. The NAM pattern is identified as the leading EOF pattern of the geopotential height at specific pressure level (here 500 hPa) and the NAM-index is then the principal component of the leading EOF. The 500 hPa level was chosen to test if there is any impact on tropospheric levels, or if impacts are limited to the middle atmosphere only. The bars in the graph presented the original distribution of the individual simulated years NAM indices (normalised values), and the smoothed lined is as described above. Same method for presenting a smoothed NAO index, for the same model, has presented by Smith, K. L., R. R. Neely, D. R. Marsh, and L. M. Polvani (2014), The Specified Chemistry Whole Atmosphere Community Climate Model (SC-WACCM), *J. Adv. Model. Earth Syst.*, 6, 883–901, doi:10.1002/2014MS000346.

We have added more information about the NAM calculation method into the Methods-section, where we now additionally write
NAM is identified as the leading EOF mode and the NAM-index is provided by the principal component of this leading model.

Comment: Figure 3, Middle. "zonal wind contour levels at 850 hPa from the REF simulation, an indication for the locations of the tropospheric jet" – How can the tropospheric jet be located at 850 hPa when the mean tropospheric jet pressure level is about 200 hPa? Should be clarified.

Reply: Our apologies for this not being clear, this part of the analysis looks at the eddy-driven jet, which is located in the 850 hPa level. We have clarified the text across to distinguish between the tropospheric jet (at 200 hPa as seen in our now Figure 2) and the eddy-driven tropospheric jet (at 850 hPa).

Comment: The discussion at Lines 169–191 needs better argumentation and clarification. For example, Lines 176–177 "The warming of up to about 1.5 K in the Northern Eurasian sector" – what is "Northern Eurasian sector"? Actually, this warming area covers areas from Scandinavia to West Siberia, but not entire Eurasia. Need to be more specific. Line 179, "and warming (up to 2 K) over North America" – actually, warming is seen over part of US, but not over the whole North America that extended from Canada to Panama. Note that there is a significant cooling (about –1.2 K) over north Canada, which is not discussed.

Reply: We have improved the text here as requested and reference to more precise geographical regions. We have also added references to previous works where our findings point to similar changes are reported earlier in reanalysis variables or in the NAM index.

We now write (note that addition of the requested diagram has shifted previous Figure 3 to Figure 4): *While the physical mechanisms have remained unverified, teleconnections between the equatorial QBO and NAM-like patterns in the Northern Hemisphere appear to be real[1]. When filtering the simulations by the QBO phase, in the REF simulation eQBO cases (Figure 4b) have a clear tendency towards negative NAM (signifying weaker polar vortex during eQBO), while the NAM index distribution is more evenly spread for wQBO (Figure 4c). In the EPPO3 simulation under eQBO conditions, there is a significant shift (at 90% level) in the peak of the distribution towards a positive NAM, which is consistent with previously found relation with geomagnetic activity (driver of EPP) and early winter NAM[2]. While there is no strengthening of the stratospheric polar vortex during eQBO, the positive NAM signal reflects the poleward shift of the tropospheric jet. For wQBO in panel d), shift towards a positive NAM is not significant; however, the poleward edge of the eddy-driven jet is strengthened (panel f). As in the unfiltered case, the central jet structure appears disrupted, with a tilt or a split. The corresponding surface level temperature differences present a positive NAM like pattern: The significant warming of up to about 1.5 K from Scandinavia to West Siberia in all cases (panels g-i) is similar to those reported previously from seasonal scale analysis[3, 2, 4]. During eQBO (panel h) there is additional significant cooling (~ -1 K) over Europe and Canada and warming (up to 2 K) over parts of United States, which is not present during the wQBO years (panel i). For wQBO, however, the high latitude Arctic in the Pacific sector cools by up to about 1 K. These patterns are broadly consistent with the significant patterns found in reanalysis temperatures in previous studies[2, 5].*

Comment: How can the time scale of the EPP influence on troposphere temperature changes be retrieved if authors compare data in two consecutive months, November and December when the source is “switched on” for four months (November–February)? The authors discuss the effect in the second month in a time when the source is still “switched on”? The physical base of the experiment needs to be clarified. What purpose do you need to reduce ozone for four months if you consider just two first months? It would be reasonable to reduce ozone for one month (November) and look for effect next month in December. Or look at the effect in March and April, when the ozone reduction is switched off.

Reply: When we designed the experiment we were not expecting to see changes in the troposphere take place so rapidly. Thus the ozone loss was determined to be switched on for the longer duration. Once results were available, it became clear that the tropospheric changes occurred already by December. In our view this rapid downward coupling was the key result to report as it can have implications for potential gain in predictive skill from inclusion of mesospheric variability. In addition, at the early winter time, the so-called indirect effects arising from downward transport of EPP produced NO_x are not yet active, which would not be the case when analysing the following months (February in particular).

Comment: It is worth more clearly explaining what physical process is modeled by the permanent mesospheric ozone decrease during four months each year. The characteristic time of short term EPP events is equal to days, and long-term modulation of the mesospheric ozone concentration happens with the solar cycle period near to 11 years.

Reply: We have now added more detailed motivation, particularly on the focus of understanding how the reported roughly 30% upper mesospheric ozone loss (over solar cycle timescales, i.e. the loss is focused on solar maximum and the declining phase of the solar-cycle), rather than looking at short term variability, which could amount to much larger ozone losses.

We now write in the Experiment design section: *To investigate the dynamical processes associated with EPP driven ozone loss in the upper mesosphere (as reported over solar-cycle timescales[6]), two sets of simulations were performed for 99 model years each: A modified*

simulation, “EPPO3”, where the Northern Hemisphere polar (60°N-90°N) mesospheric (0.01-0.05 hPa) ozone levels were reduced by 30 % from the SC-WACCM background levels during northern winter months (November-February). This corresponds to reported observations[6] of EPP impact on polar ozone over roughly monthly timescales. The ozone depletion results in a 0.5 K/day increase in long wave heating in the EPPO3 simulation, consistent with other experiments[7]. Ozone loss at these pressure levels would correspond to energetic electron precipitation with energies in the range of few hundred keV, or proton precipitation few MeV to about 10 MeV energies[8]. For comparison, the second simulation set, “REF”, was performed identically to EPPO3, but without the ozone modification. As we are solely focused on investigating the dynamical implications of controlled ozone depletion, these model simulations enable us to account for internal variability and isolate the effects arising from the polar mesospheric ozone loss. We note that larger ozone depletion has been reported over shorter timescales[6], but our focus here is solely to investigate processes relating to dynamical signals that have been reported at monthly to seasonal scales (relating to solar cycle timescales in EPP variability).

Technical comment:

Comment: Lines 244-246. Two authors do not have any specific function.

Reply: This was unintentional and has been fixed. Both authors (AYK and MES) had specific roles which we now explain: AYK provided expertise on stratosphere-troposphere coupling and MES contributed to the model simulations.

Comment: Finally, what are the primary and new results? Is there a change in tropospheric temperature by 1–2K in some regions (bottom panel of Fig. 3)? It should be more explicit and emphasized in the text. Generally, authors proposed a fruitful hypothesis when the early winter (November) process of the polar mesospheric ozone reduction under EPP impact produces the chain of signals propagating from the mesosphere through the stratosphere to the lower tropospheric altitudes in mid-latitudes. However, this hypothesis still needs a more detailed explanation of the processes chain and modeling results that authors can provide. That can help increase the significance of the results obtained, which is really important to better understanding EPP-mesosphere-stratosphere-troposphere coupling processes.

Reply: Our primary result is reporting on the dynamical coupling which is able to transfer the mesospheric ozone loss signal to the troposphere at timescales that correspond to previously reported surface level changes. To highlight the role of the coupling outside of the polar region (mid- and low latitudes) we have added further discussion on this the first paragraphs of the manuscript. We have also clarified that we are solely focused on looking at the effects flowing on from in situ ozone loss in the mesosphere, and not the EPP-indirect effect, which would require a carefully designed separate model study. We also now include a diagram illustrating the processes we have identified as requested (new Figure 1).

Bibliography

- [1] Andrews, M. B. *et al.* Observed and Simulated Teleconnections Between the Stratospheric Quasi-Biennial Oscillation and Northern Hemisphere Winter Atmospheric Circulation. *J. Geophys. Res.: Atmos.* **124**, 1219–1232 (2019).
- [2] Seppälä, A., Randall, C. E., Clilverd, M. A., Rozanov, E. V. & Rodger, C. J. Geomagnetic activity and polar surface air temperature variability. *J. Geophys. Res.: Atmos.* **114**, A10312 (2009).
- [3] Rozanov, E. V. *et al.* Atmospheric response to NO_y source due to energetic electron precipitation. *Geophys. Res. Lett.* **32**, L14811 (2005).
- [4] Arsenovic, P. *et al.* The influence of Middle Range Energy Electrons on atmospheric chemistry and regional climate. *J. Atmos. Sol. Terr. Phys.* **149**, 180–190 (2016).
- [5] Maliniemi, V., Asikainen, T., Mursula, K. & Seppälä, A. QBO-dependent relation between electron precipitation and wintertime surface temperature. *J. Geophys. Res.: Atmos.* 6302–6310 (2013).
- [6] Andersson, M. E., Verronen, P. T., Rodger, C. J., Clilverd, M. A. & Seppälä, A. Missing driver in the Sun-Earth connection from energetic electron precipitation impacts mesospheric ozone. *Nature Comm.* **5**, D07307 (2014).
- [7] Meraner, K. & Schmidt, H. Climate impact of idealized winter polar mesospheric and stratospheric ozone losses as caused by energetic particle precipitation. *Atmos. Chem. Phys.* **18**, 1079–1089 (2018).
- [8] Matthes, K. *et al.* Solar forcing for CMIP6 (v3.2). *Geoscientific Model Development* **10**, 2247–2302 (2017).

Author responses to reviewer comments on paper “Polar mesospheric ozone loss initiates downward coupling of solar signal in the Northern Hemisphere”.

Our detailed responses to comments from Reviewer # 1 are included here. We note that the other reviewers did not have further comments to address.

Comments from Reviewer # 1

Comment: The authors of the manuscript “Polar mesospheric ozone loss initiates downward coupling of solar signal in the Northern Hemisphere” base their theory on tropospheric temperature changes in December, suggesting a dynamic coupling of the mesosphere, stratosphere, and troposphere that began with a 30% reduction in mesospheric ozone due to EPP forcing in November.

The authors strongly favor the term “energetic particle precipitation (EPP)” but cite only one paper that estimates % ozone depletion by “energetic electron precipitation (EEP)”. [Andersson, M. E., Verronen, P. T., Rodger, C. J., Clilverd, M. A. & Seppälä, A. Missing driver in the Sun-Earth connection from energetic electron precipitation impacts mesospheric ozone. *Nature Comm.* 5, D07307 (2014).]

If authors want to use the phrase energetic particle precipitation, they should add some references that estimate % ozone depletion by energetic proton precipitation (during solar energetic particle events (SEP)). Many studies have reported a decrease in mesospheric ozone of approximately several tens of percent associated with separate SEPs [e.g. Weeks et al., 1972; McPeters et al., 1981; Thomas et al., 1983; Jackamn et al., 2008; etc.] and with SEPs over solar cycles [McPeters and Jackman, 1985; Rozanov et al., 2012 Doronin et al., 2024]

Please add at least the suggested references: [McPeters, R. D., and C. H. Jackman (1985), The response of ozone to solar proton events during solar cycle 21: The observations, *J. Geophys. Res.*, 90(D5), 7945–7954, doi:10.1029/JD090iD05p07945.]

[Rozanov, E., Calisto, M., Egorova, T. et al. (2012), Influence of the Precipitating Energetic Particles on Atmospheric Chemistry and Climate. *Surv Geophys* 33, 483–501. <https://doi.org/10.1007/s10712-012-9192-0>]

[Doronin, G.; Mironova, I.; Bobrov, N.; Rozanov, E. (2024), Mesospheric Ozone Depletion during 2004–2024 as a Function of Solar Proton Events Intensity. *Atmosphere*, 15, 944. <https://doi.org/10.3390/atmos15080944>]

in the text of the manuscript: Page 1, 6th line from the bottom “ozone changes driven by EPP on solar cycle timescales” Page 3, 2nd line from the bottom “with >30 % ozone variation on solar cycle time scales” Page 10, 2nd line from top “EPP driven ozone loss in the upper mesosphere (as reported over solar-cycle timescales)”

Reply: Reference #1: We unfortunately had to remove all references from the abstract, so instead we have added this reference to the Introduction where this matter is discussed.

Reference #2: We already cited this work earlier in the article, and have now added an additional reference to the requested location.

Reference #3: This article is not relevant to the work here. As we already include other references relating to Solar Proton Events (including reference #1 as pointed out by the reviewer), we have chosen not to include this one.

Comment: It is hard to believe that the mesospheric ozone layer depletes by 30% each November over several solar cycles. However, proving this is not the purpose of this paper. But it should be explained correctly that the authors are using temperature changes in December based only on the EPP forcing during November. Include changes in the phrase “by 30% from the SC-WACCM

background levels during northern winter months (November-February)”, clarifying that the authors mean, “by 30% the SC-WACCM background levels during each northern winter month”.

Reply: We have added the additional word “each” to the text as requested.